# Effect of Single High Dose Vitamin D Substitution in Hospitalized COVID-19 Patients with Vitamin D Deficiency on Length of Hospital Stay

**DOI:** 10.3390/biomedicines11051277

**Published:** 2023-04-25

**Authors:** Fabienne Jaun, Maria Boesing, Giorgia Luethi-Corridori, Kristin Abig, Nando Bloch, Stéphanie Giezendanner, Victoria Grillmayr, Philippe Haas, Anne B. Leuppi-Taegtmeyer, Jürgen Muser, Andrea Raess, Philipp Schuetz, Michael Brändle, Jörg D. Leuppi

**Affiliations:** 1University Center of Internal Medicine, Cantonal Hospital Baselland, 4410 Liestal, Switzerland; maria.boesing@ksbl.ch (M.B.); giorga.luethi-corridori@ksbl.ch (G.L.-C.); kristin.abig@ksbl.ch (K.A.); philippehaas@mac.com (P.H.); anne.leuppi-taegtmeyer@usb.ch (A.B.L.-T.); 2Medical Faculty, University of Basel, 4001 Basel, Switzerland; philipp.schuetz@ksa.ch; 3Paracelsus Medical University, Master Programme Public Health, Center for Public Health and Healthcare Research, 5020 Salzburg, Austria; 4Cantonal Hospital St. Gallen, 9000 St. Gallen, Switzerland; nando.bloch@kssg.ch (N.B.); victoria.grillmayr@kssg.ch (V.G.); andrea.raess@kssg.ch (A.R.); michael.brandle@kssg.ch (M.B.); 5Central Laboratories, Cantonal Hospital Baselland, 4410 Liestal, Switzerland; 6Cantonal Hospital Aarau, 5001 Aarau, Switzerland

**Keywords:** vitamin D, COVID-19, SARS-CoV-2, vitamin D deficiency, vitamin D substitution, immunomodulation

## Abstract

Vitamin D and its role in the coronavirus-19 disease (COVID-19) pandemic has been controversially discussed, with inconclusive evidence about vitamin D3 (cholecalciferol) supplementation in COVID-19 patients. Vitamin D metabolites play an important role in the initiation of the immune response and can be an easily modifiable risk factor in 25-hydroxyvitamin D_3_ (25(OH)D_3_)-deficient patients. This is a multicenter, randomized, placebo-controlled double-blind trial to compare the effect of a single high dose of vitamin D_3_ followed by treatment as usual (TAU) of daily vitamin D_3_ daily until discharge versus placebo plus TAU in hospitalized patients with COVID-19 and 25(OH)D_3_-deficiency on length hospital stay. We included 40 patients per group and did not observe a significant difference in the median length of hospital stay (6 days in both groups, *p* = 0.920). We adjusted the length of stay for COVID-19 risk factors (β = 0.44; 95% CI: −2.17–2.22), and center (β = 0.74; 95% CI: −1.25–2.73). The subgroup analysis in patients with severe 25(OH)D_3_-deficiency (<25 nmol/L) showed a non-significant reduction in the median length of hospital stay in the intervention group (5.5 vs. 9 days, *p* = 0.299). The competing risk model with death did not reveal significant differences between the group in the length of stay (HR = 0.96, 95% CI 0.62–1.48, *p* = 0.850). Serum 25(OH)D_3_ level increased significantly in the intervention group (mean change in nmol/L; intervention: +26.35 vs. control: –2.73, *p* < 0.001). The intervention with 140,000 IU vitamin D_3_ + TAU did not significantly shorten the length of hospital stay but was effective and safe for the elevation of serum 25(OH)D_3_ levels.

## 1. Introduction

For more than two years, the world has been facing a global pandemic, with a then new coronavirus (CoV). The underlying pathogen of this new disease was later identified as Severe Acute Respiratory Syndrome Coronavirus-2 (SARS-CoV-2) [1]. Within a short time period, the virus spread quickly around the world, and led the World Health Organization (WHO) to reclassify the epidemic as a global pandemic in March 2020 [2]. 

The symptoms that occur in a COVID-19 infection are similar to those shown in patients with SARS or MERS, which is explained by the genetic relationship of the virus [3]. Risk factors for a poor outcome of COVID-19 have been found to include older age, male sex, obesity, chronic respiratory conditions, cardiovascular disease (CVD), metabolic diseases, malignancies, and active smoking [4,5,6,7,8]. 

There are studies that support the theory that 25(OH)D_3_ deficiency plays a role in the progression and severity of COVID-19, other studies could not find a significant connection between the two conditions [9]. However, the scientific community has not yet found a consensus about the role of vitamin D metabolites in the current pandemic. This randomized, placebo-controlled, double-blind trial investigated if a single high dose of vitamin D_3_, in addition to standard dosed vitamin D_3_ supplementation, can positively influence the length of hospital stay in patients with 25(OH)D_3_ deficiency and COVID-19 compared to patients receiving the treatment as usual for 25(OH)D_3_ deficiency only. 

Deficiency of 25(OH)D_3_ is a well-documented phenomenon that occurs in approximately 40% to 60% of the population, depending on whether the threshold is 50 nmol/L or 75 nmol/L as defining criteria for deficiency [10]. There are observational studies reporting independent associations between low serum concentrations of 25(OH)D_3_ and susceptibility to acute respiratory tract infection [11,12]. 25(OH)D_3_ supports the induction of antimicrobial peptides in response to both viral and bacterial stimuli, suggesting a potential mechanism by which vitamin D inducible protection against respiratory pathogens might be mediated [13,14,15]. 

Various studies have been conducted that investigated the influence and role of vitamin D_3_ supplementation in various dosages (from medium dose to ultra-high dose) in patients with critical illnesses. A retrospective observational study of the serum 25(OH)D_3_ status in patients with ARDS, which is a common complication in patients with COVID-19, showed that over 95% of these patients had 25(OH)D_3_ deficiency (*n* = 108) [16]. So far, the influence of vitamin D_3_ in the pandemic is unclear, the current level of evidence is low and contradictory. This might be influenced by various factors, such as the lack of high-quality evidence, different methodological approaches, or inconsistent enrolment criteria. Some studies found significant differences on various variables, such as the time to resolve symptoms or biomarker, mortality, or ICU admission [17,18,19,20,21], whereas others did not find any relationship between those outcomes and vitamin D_3_ supplementation [22,23,24]. Most of the studies or reviews concluded that further research in the form of randomized controlled trials is indicated to clarify the impact of vitamin D_3_ supplementation in patients with COVID-19, as there are still unanswered questions and uncertainties that need clarification. Previous findings in vitamin D research found a reduction in ventilation days for critically ill patients, a better muscle tissue function, and suppressed cytokines in patients who were treated with high dose vitamin D_3_ [25]. According to previous studies, there was no increased risk for adverse events or side effects observed when patients were treated with high dose vitamin D [26,27]. Further, there are observations that a single high dose of vitamin D is more effective in elevating the serum 25(OH)D_3_ levels, whereas daily administration is more beneficial for health outcomes [28]. We therefore came to hypothesize that a single high dose of vitamin D_3_ in addition to standard treatment improves the recovery period positively in patients with COVID-19 and 25(OH)D_3_ deficiency compared to daily smaller doses of vitamin D_3_ only. That means that the time to recovery, which was defined as length of hospital stay, is shorter in the single high dose vitamin D_3_ group relative to standard treatment group only.

## 2. Materials and Methods

This is a multicenter, randomized, placebo-controlled double-blind trial to compare the effect of a single high dose of vitamin D_3_ (140,000 IU, Vitamin D3 Wild Oel 500 IU/drop, Muttenz, Switzerland ) followed by treatment as usual (TAU) of daily vitamin D_3_ (800 IU, Vitamin D3 Streuli 4000 IU/ml, Streuli Pharma AG, Uznach, Switzerland) until discharge versus placebo plus TAU in hospitalized patients with COVID-19 and 25(OH)D_3_ deficiency on length hospital stay. All patients that were hospitalized due to COVID-19 in one of the participating centers (Cantonal hospital Baselland and Cantonal hospital St. Gallen) were considered as potentially eligible for participation in this trial and therefore checked for the inclusion and exclusion criteria (Table 1). The patients were recruited between December 2020 and August 2021. Participants were randomized in a 1:1 ratio to either group and randomization was stratified by center using block randomization (block size = 4). 

To elevate the serum 25(OH)D_3_ level quickly and sufficiently, a single dose of 140,000 IU (3500 μg) vitamin D_3_ was administered orally. This dose was found to be safe and effective in a study of high-dose vitamin D_3_ in patients with tuberculosis [29,30]. The dosage recommended for the standard treatment of a vitamin D_3_ deficiency varies from 400 to 1000 IU per day [31]. In consultation with a pharmacovigilance expert (University Hospital of Basel), the study team considered that the dose of 800 IU was adequate to elevate the serum 25(OH)D_3_ level in the included patients. 

### 2.1. Outcomes

The primary outcome was defined as the length of the hospital stay (LoS), defined as the time from randomization to discharge. Further, the time from symptom onset to discharge and the time from hospital admission to discharge were assessed and analyzed. 

As secondary outcomes, we defined COVID-19-related complications, namely ICU admission, mechanical ventilation, death, and other complications linked to COVID-19. Additionally, we looked at the serum 25(OH)D_3_ levels, the health-related quality of life on days 28 and 90, self-reported symptoms, and vital signs. For safety outcomes, the occurrence of adverse events (AEs) and serious adverse events (SAEs) were assessed [32]. 

### 2.2. Data Collection and Management

Each participant underwent the same procedure regarding the study-related visits from inclusion until discharge and for the follow-up period (Appendix A). The patients had daily visits until discharge by treating physicians and nurses. The Short Form 12 item health survey (SF-12) to assess the health=related quality of life (hrQoL) on day 28 and day 90 days after randomization was assessed by a telephone visit. The data were collected in an electronic data capture system (SecuTrial^®^), which is accessible via a standard browser and personalized and password protected. 

### 2.3. Statistical Analysis

To perform the statistical analysis, either R Statistical software (Version 4.0.3) or SPSS Statistics, (Version 24 (IBM)) was used. We performed a sample size calculation based on the assumption of a true difference in the mean length of stay of two days (SD = 2.96) [33]. According to the calculation, accounting for 10% dropouts and the chosen block size randomization, a total number of 80 patients (40 per group) were included. 

For the primary outcome, we used the Mann–Whitney-U, due to non-normal distribution, unless our assumptions for the sample size calculations were based on a two-sided *t*-test. For continuous data, independent two-sample Student *t*-test or Mann–Whitney-U test was used for secondary analyses of group differences in the variables of interest. To compare the proportion of patients between the two groups regarding the secondary outcomes, a Chi^2^ test was performed and if values had expected frequencies <5, the Fisher exact test was used with simulated p-value (based on 2000 replicates). To evaluate the change in the SF-12 questionnaire over three time points, linear mixed effects models were used to determine time-by-treatment interactions using the Satterthwaite approximation to estimate *p*-values. Sensitivity analysis of the primary outcome was performed with a robust linear regression model adjusted for known COVID-19 risk factors. To evaluate the safety of short-term high dose vitamin D_3_ administration in the above-described population, linear/logistic mixed-effect models were used to estimate treatment effects on adverse and serious adverse events across time.

A competing risk analysis that accounts for death as competing risk (besides the event of interest which was time to discharge) using the Fine and Gray model and a subgroup analysis for patients who had severe vitamin D deficiency at randomization, defined as 25(OH)D_3_ level <25 nmol/L, for the primary endpoint with the same method used in the full dataset was performed post hoc. We only performed the binary analysis for the subgroup based on the small number of patients in the subgroup. The detailed methodology of the trial has been published previously [34]. 

## 3. Results

A total of 80 patients were randomized, 40 patients in each group. Two patients were excluded after randomization because the serum 25(OH)D_3_ level at the day of inclusion was >50 nmol/L. The data of 78 patients were used for the analysis. A total of 39 patients were in the intervention group and 39 patients in the placebo group. Median time from hospital admission to randomization was two days for both groups (SD ± 1 day).

### 3.1. Clinical Characteristics at Time of Randomization

The baseline characteristics of our population are summarized in Table 2. The mean age of our population was 60.5 (range 35–91) years in the intervention group and 61.5 (range 29–93) years in the control group. There were more women in the intervention group (35.9%) compared to the control group (17.9%; *p* = 0.125). Mean 25(OH)D_3_ level at time of randomization was 31.03 (SD ± 10.95) nmol/L in the intervention group and 28.54 (±10.13) nmol/L for the control group (*p* = 0.232). The two groups did not differ in terms of mean vital signs on the day of randomization. The mean duration from symptom onset to hospital admission was 8 days (range 3–18) in the intervention group versus 7 days (range 1–19) in the control group (*p* = 0.335). Almost all patients (intervention: 100% vs. control: 95%) showed COVID-19 typical changes in the radiological imaging of the lungs (*p* = 0.152). The intervention group had a significantly higher mean CRP value at the time of randomization compared to the control group (intervention: 96.50 mg/L vs. control: 66.67 mg/L, *p* = 0.015); no other value in the hematology profile or blood chemistry showed a significant difference between the groups. The mean SF-12 mental score (hrQoL sub score) was significantly higher in the control group compared to the intervention group (intervention: 41.76 ± 15.51 versus control: 51.66 ± 8.04; *p* = 0.018). In terms of co-morbidities at the time of randomization and concomitant medication during the hospitalization, no significant differences between the groups were observed (Appendix A). 

### 3.2. Primary Outcome

There was no significant difference between intervention group and control group in terms of length of hospital stay (Table 3, Figure 1). Median time from symptom onset to discharge was 16 days for the intervention group and 15 days for the control group (*p* = 0.193). Median time from hospital admission to discharge was 8 days for both groups (*p* = 0.924) and the median time from randomization to discharge was 6 days for both groups (*p* = 0.920). 

We adjusted the length of stay variables for COVID-19 risk factors, including age, number of comorbidities, peripheral oxygen saturation, number of symptoms, and 25(OH)D_3_ value at randomization (β = 0.03, 95% CI: −2.17–2.22), prognostic imbalances between the groups (adjusted for *p* < 0.010) (β = 0.39, 95% CI: −1.78–2.56), and center (β = 0.74, 95% CI: −1.25–2.73) (Appendix A). 

#### Post Hoc Analysis

The post-hoc analysis of patients with severe 25(OH)D_3_ deficiency showed a median time from symptom onset to discharge of 15 days in both groups (*p* = 0.764). Median time from hospital admission to discharge was 5.5 days for the intervention group and 9 days for the control group (*p* = 0.299) and the median time from randomization to discharge was 4.5 days for the intervention group and 7 days for the control group (*p* = 0.444) (Table 4). 

In the competing risks model with the full dataset and death as the competing risk, groups did not differ in (a) time from symptom onset to discharge (HR = 0.88, 95% CI 0.57–1.36, *p* = 0.570), (b) time from hospital admission to discharge (HR = 0.99, 95% CI 0.64–1.52, *p* = 0.950), or (c) time from randomization to discharge (HR = 0.96, 95% CI 0.62–1.48, *p* = 0.850) Figure 2.

### 3.3. Secondary Outcomes

In each group, four patients were admitted to ICU (intervention: 10.3% (n = 4) vs. control: 10.3% (n = 4). The median time treated on ICU was significantly longer in the intervention group, compared to the control group (intervention: 14 days (13–17) vs. control: 6 days (1–13), *p* = 0.028). As one patient was not admitted due to COVID-19, we also performed a corrected comparison, which included only patients who were admitted to ICU due to COVID-19 (Table 5)). 

In total, five patients were intubated, without a difference between the groups (intervention: 10.3% (n = 4) vs. control: 2.6% (n = 1); OR 4.343, 95% CI: 0.463–40.749). Overall, three participants died during this trial (intervention: 2.6% (n = 1) vs. control: 5.1% (n = 2), *p* = 1.00; OR 0.487, 95 CI: 0.042–5.601). Other COVID-19-related complications were reported in 5 out of 78 patients (intervention: 7.7% (n = 3) vs. control: 5.1% (n = 2), *p* = 1.00; OR 1.542, 95 CI: 0.243–9.776) (Table 4). 

At randomization, the two groups did not differ in the mean serum 25(OH)D_3_ level (intervention: 31.46 nmol/L (SD ± 10.95) (range: 11–44) vs. control: 28.54 nmol/L (SD ± 10.13) (range: 9–45); *p* = 0.232). At discharge, the intervention group had higher mean serum 25(OH)D_3_ D level compared to the control group (intervention: 50.95 nmol/L (SD ± 16.98) (range: 20–83) vs. control: 26.35 nmol/L (SD ± 8.88) (range: 9–45); *p* < 0.001). The mean increase in serum 25(OH)D_3_ was higher in the intervention group, whereas the mean 25(OH)D_3_ level in the control group decreased (intervention: +22.81 nmol/L (SD ± 15.23) (range: −5–48) vs. control: −2.73 nmol/L (SD ± 10.23) (range: −33–16); *p* < 0.001). (Table 6, Figure 3).

At randomization, there was a significant difference in the mean SF-12 mental score between the groups (intervention: 41.759 ± 15.510 vs. control: 51.656 ± 8.044, *p* = 0.018). No significant difference in the mean SF-12 physical score was observable at randomization (intervention: 31.39 ± 10.61 vs. control: 27.85 ± 11.75, *p* = 0.352). In the first follow-up call 28 days after inclusion, there was no more significant difference in the mean SF-12 mental score between the groups observed (intervention: 51.09 ± 13.57 vs. control: 51.86 ± 8.82, *p* = 0.775). The mean SF-12 physical score did not differ significantly at the first follow-up (intervention: 38.73 ± 9.14 vs. control: 43.41 ± 10.35, *p* = 0.136). In the second follow-up call 90 days after inclusion, again, no significant difference in the mean SF-12 mental score (intervention: 47.58 ± 11.79 vs. control: 52.81 ± 9.49, *p* = 0.136) or the mean SF-12 physical score (intervention: 45.07 ± 11.72 vs. control: 50.81 ± 5.70, *p* = 0.136) was observed ( Appendix A). The linear mixed effect model showed a significant time-by-treatment interaction for SF-12 physical evaluation (*p* = 0.023), indicating that the intervention group had a less pronounced increase of the SF-12 physical scores across time (−4.56 per measured time point). On the SF-12 mental score evaluation, no significant time-by-treatment interaction was observed (*p* = 0.272, 2.75 per measured time point) (Appendix A).

On day five after randomization, it was observed that significantly more patients reported headache within the intervention group compared to the control group (intervention: 56.3% vs. control: 15%, *p* = 0.014) (Appendix A). There was no other significant difference in self-reported symptoms observable at day 5 after randomization. The groups did not differ in terms of vital signs (Appendix A). Ten days after randomization, there were no more significant differences in terms of self-reported symptoms observed between the groups. In terms of vital signs, the intervention group had a lower systolic blood pressure (*p* = 0.029), but no difference was observed in other vital signs (Appendix A). 

Non-significantly fewer adverse events were reported in the intervention group compared to the control group (intervention: 10.3% vs. control: 25.6%, *p* = 0.138). For serious adverse events, the intervention group had non-significantly more SAEs reported compared to the control group (intervention: 10.3% vs. control: 5.1%, *p* = 0.675) (Appendix A). We did not observe a significantly higher mean calcium level in the intervention group compared to the control group (intervention: 2.30 mmol/L (SD ± 0.99) vs. control: 2.22 mmol/L (SD ± 0.063); *p*= 0.101). 

We observed a significant higher mean PTH level in the control group compared to the intervention group (intervention: 2.98 pmol/L (SD ± 1.30) vs. control: 1.60 pmol/L (SD ± 2.16); *p* = 0.007) and a significant lower mean level of phosphate in the control group compared to the intervention group (intervention: 1.01 mmol/lL(SD ± 0.289) vs. control: 0.86 mmol/L (SD ± 2.66); *p* = 0.003) (Appendix A).

## 4. Discussion

We could not observe a difference in the primary endpoint length of hospital stay. The comparison of the length of stay variables between the two groups did not reveal a significant difference that would indicate a beneficial effect of a single high dose vitamin D_3_ in addition to standard treatment. The findings of our study are in line with previous results [17,18,22,23,24,35]. The median length of stay in our population was similar to the findings presented by Murai et al. (2021) [22]. 

In the post-hoc subgroup analysis, which only included patients with a 25(OH)D_3_ deficiency (<25 nmol/L), we observed a non-significant shorter length of stay for the intervention group. There was no difference in the median time from symptom onset to discharge, but in the other length of stay variables (“hospital admission to discharge” and “randomization to discharge”). This trend was already observed by other studies, therefore it might be worth having a closer look at patients with severe 25(OH)D_3_ deficiency in bigger studies, as our sub-group was underpowered to detect a clinically significant difference [18,36].

In terms of secondary outcomes, we could not find significant differences in the need of ICU treatment, intubation, death, or other COVID-19 related complications. Those findings indicate that high dose vitamin D_3_ had no impact on the course of the disease regarding severity. We observed a significantly longer mean duration in the ICU within the intervention group. In the overall population, there were fewer admissions to the ICU compared to other studies, which was possibly influenced by the fact that our population had fewer co-morbidities and therefore the risk of ICU admission was smaller [22,37]. With our study, we could not support previous findings that high dose vitamin D_3_ supplementation reduces ICU admissions [37]. The fact that more patients in the intervention group were intubated compared to the control was not significant and we therefore assume that individual predisposing factors or pre-existing conditions have had a greater influence than the intervention with high doses of vitamin D_3_ in patients with 25(OH)D_3_ deficiency [5]. This might also contribute to the significantly longer length of ICU treatment in the intervention group. Our study could not confirm the results that fewer patients needed ICU admission or less mechanical ventilation when treated with high dose vitamin D_3_ in addition to the best available other therapies [37,38,39]. However, the number of patients with COVID-19-related complications was very small and therefore the results need to be interpreted with the utmost caution. 

We observed a significant increase in the mean serum 25(OH)D_3_ level in the intervention group compared to the control group, which resulted in a significant higher mean level at discharge in the intervention group and significantly more patients had serum 25(OH)D_3_ levels ≥50 nmol/L in the intervention group. At the same time, there was no clinically relevant increase in calcium or phosphate in the intervention group, which is in line with previous results [24,40,41,42]. This indicates that a single bolus of high dose vitamin D_3_ in addition to daily low dose maintenance therapy is effective to elevate serum 25(OH)D_3_ levels quickly. This finding is in line with previous observations that investigated the effect of a single high dose vitamin D_3_ with or without daily maintenance dose in various conditions [29,43,44,45]. Another factor that might have influenced the outcome of our study is the individual response to vitamin D_3_ supplementation, as only 23.1% of the participants in the intervention group reached sufficient 25(OH)D_3_ serum levels [46].

At time of inclusion, we observed significantly lower SF-12 mental scores in the intervention group than in the control group. This might have led to a bias in the primary outcome variable, since higher hrQoL scores are associated with a reduction of LOHS [47]. The difference between the groups in the SF-12 mental sub-score was not observable anymore 28 days and 90 days after randomization. The increasing number of missing data over time might mediate this observation. Another aspect that should be considered in the discussion of the observation is the range of the SF-12 mental score in our population. In the intervention group, both the range and the variance were larger than in the control group. However, the sample with a completed hrQoL from baseline to day 90 was rather small and therefore possible effects of the two groups did not differ significantly in terms of self-reported symptoms or recorded vital signs. At day five after randomization, we observed that a significantly higher number of patients in the intervention group reported headache compared to the control group, but no differences in other symptoms or vital signs. Ten days after randomization, no more differences were observed in self-reported symptoms. When looking at the vital signs, we observed a significantly lower systolic blood pressure in the intervention group compared to control group, but still the values were in the normal range. Orthostatic hypotension can be caused by longer bedrest or infections [48,49]. Further, only 10 and 11 patients respectively were still hospitalized at day 10 after randomization and therefore the results must be interpreted with caution. Our study could not confirm the findings which reported faster symptom recovery [17,18]. 

Our results did not show significant differences in the occurrence of adverse and serious adverse events between the intervention group and the control group. This indicates that the intervention with high dose vitamin D_3_ is safe and not associated with more adverse or serious adverse events. 

Our study population showed similar characteristics at the time of randomization such as age, median time from symptom onset to hospitalization, laboratory parameters, or 25(OH)D_3_ level compared to other trials, except the sex distribution, as our study had fewer female participants [17,18,21,22,23,24,35,40,41,50,51,52,53]. In the early stage of the pandemic in Switzerland, it was observed that the majority of hospitalized patients were male (female: 37% vs. male: 63%), which might explain the lower number of females in our study [54]. In terms of hematological and blood chemistry parameters, we observed several values that were above or beyond the normal value range, which can be explained by viral infections, including SARS-CoV-2, medication, systemic disorders, or malnutrition [55,56,57,58,59,60,61,62]. The intervention group showed significantly higher baseline CRP values than the control group. This difference might imply that patients in the intervention group were suffering from a more severe disease compared to the control group, as a higher CRP could be an early sign for more severe COVID-19 [63,64]. However, since there were no significant differences in baseline respiratory status suggesting a more severe disease in the intervention group (respiratory rate, SpO2, dyspnea, O2-supplementation), the authors cannot conclude if this imbalance had an effect on the overall results of this study.

Our population had a lower frequency of comorbidities reported compared to other studies, namely arterial hypertension and other CVDs [22,24,35,53,65,66], which might influenced the overall outcome. 

The time between symptom onset and randomization was rather long with more than 7 days in both groups, which might have had a influence on the primary outcome, as already reported by Murai and colleagues [22]. 

### Strengths and Limitations

Among the strengths of our study are the study design as a randomized, controlled double blind multicenter trial, that the groups did not differ significantly across demographic and clinical variables at time of randomization, and the fact that all patients completed the trial. Limitations for this trial were the small sample size of 80 patients included and 78 analyzed, and missing values in some laboratory parameters at randomization and discharge and the health-related quality of life in the follow-up period. Further, we did not assess COVID-19 severity with a validated scale. In addition, we did not include outpatients in this trial and the period between symptom onset and randomization was rather long. 

## 5. Conclusions

In this trial, we could not confirm previous findings that a bolus of high dose vitamin D_3_ had a beneficial effect on the investigated outcomes patients with COVID-19 and 25(OH)D_3_ deficiency, except for the elevation of the serum 25(OH)D_3_ level. In the subgroup analysis, we observed a non-significant difference in median LOS, which might indicate that patients with severe 25(OH)D_3_ deficiency would benefit more from the high dose intervention compared to the overall 25(OH)D_3_ deficient population in our study [36,67,68]. Whether this finding was due to the beneficial effect of vitamin D_3_ or due to chance remains unclear, since the subgroup analysis was underpowered, and the assumption needs further investigation for verification. In further studies focusing on patients with severe 25(OH)D_3_ deficiency, non-hospitalized patients or patients in nursing homes should be included and the outcome of fewer hospital admissions needed should be added. We could show that a single high dose vitamin D_3_ in addition to standard vitamin D_3_ is not associated with adverse or serious adverse events, and therefore a safe and effective way to elevate the serum 25(OH)D_3_ level quickly, whereas daily small dosages of vitamin D_3_ supplementation did not elevate the serum 25(OH)D_3_ level sufficiently. 

## 6. Participant Privacy and Confidentiality

All individual medical information that was obtained during this trial is considered confidential and disclosure to third parties is prohibited. Only ethics committees or regulatory authorities are allowed to have access to source data within the framework of an audit of the study. The confidentiality is further ensured by utilizing subject identification numbers for data record in the database. 

## Figures and Tables

**Figure 1 biomedicines-11-01277-f001:**
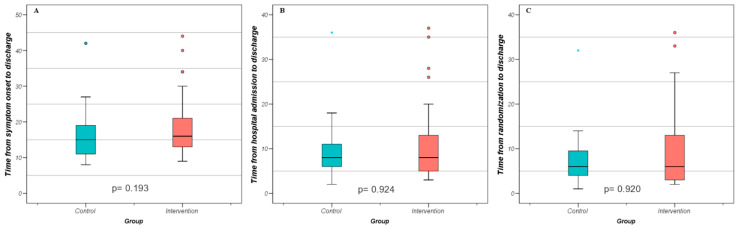
Length of hospital stay as boxplots over the variables “time from symptom onset to discharge”, “time from hospital admission to discharge”, and “time from randomization to discharge” with n = 78 (39 patients per group). (**A**). “Time from symptom onset to discharge” in days (intervention in blue and control in red), *p* = 0.193. The dots are outliers (1.5 IQR). (**B**). “Time from hospital admission to discharge” (intervention in blue and control in red), *p* = 0.924. The dots indicating the outliers (1.5 IQR). The blue star is an extreme value (3 IQR). (**C**). “Time from randomization to discharge” (intervention in blue and control in red), *p* = 0.920. The dots are outliers (1.5 IQR). The blue star is an extreme value (3 IQR).

**Figure 2 biomedicines-11-01277-f002:**
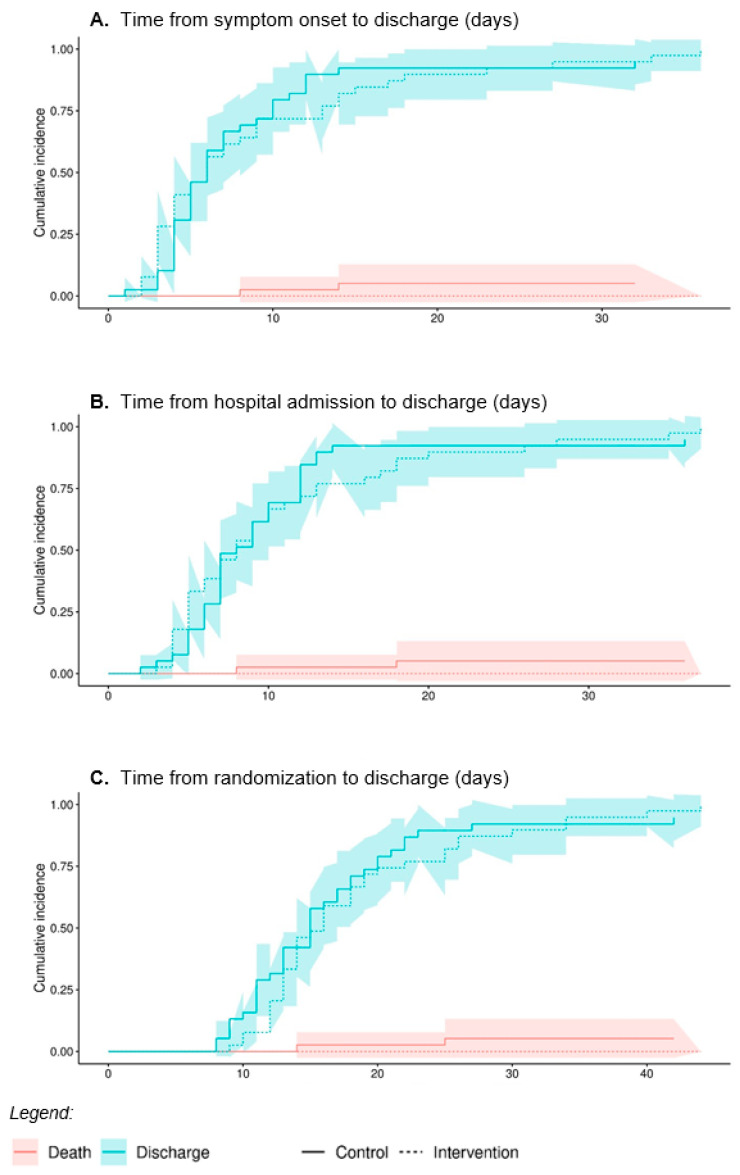
Competing risk model length of stay with death as competing risk. The blue lines indicate the event “discharge”, whereas the red indicates the competing risk “death”, with its 95% confidence interval. (**A**). Time from symptom onset to discharge, event discharge: HR 0.88 (95% CI 0.57–1.36), *p* = 0.570. Event Death: HR 0.47 (95% CI: 0.4–5.02), *p* = 0.540. (**B**). Time from hospital admission to discharge, event discharge: HR 0.99 (95% CI 0.64–1.52), *p* = 0.950. Event Death: HR 0.49 (95% CI: 0.05–5.16), *p* = 0.550. (**C**). Time from randomization to discharge, event discharge: HR 0.69 (95% CI 0.62–1.48), *p* = 0.850. Event Death: HR 0.49 (95% CI: 0.05–5.16), *p* = 0.550.

**Figure 3 biomedicines-11-01277-f003:**
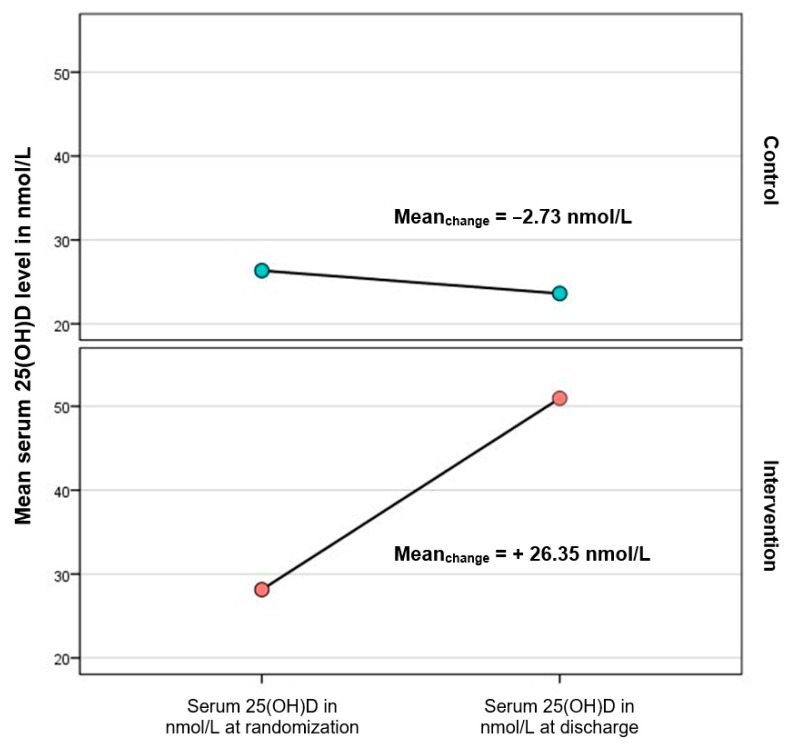
Mean change in serum 25(OH)D from time of randomization.

**Table 1 biomedicines-11-01277-t001:** Inclusion and exclusion criteria.

**Inclusion Criteria**
Informed consent as documented by signature
>18 years old
Ongoing COVID-19 infection (PCR confirmed)
Hospitalized in one of the participating study centers
Laboratory-confirmed serum 25(OH)D_3_ < 50 nmol/L (<20 ng/mL)
**Exclusion criteria**
Known hypersensitivity against one of the used vitamin D_3_ products
Active malignancy
Hypercalcemia defined as serum calcium >2.2 mmol/L
Granulomatous disease (e.g., sarcoidosis)
History of renal stones in the past 12 months
Pregnancy or breastfeeding
Admission to ICU before inclusion or at day of inclusion

**Table 2 biomedicines-11-01277-t002:** Demographic and baseline characteristics of the included patients at time of randomization.

Baseline Characteristics		N ^a^ (Intervention/Control)	Intervention	Control	*p*-Value
Age	years	39/39	60.49 ± 13.84	61.38 ± 15.29	0.787
Sex	female, % (n)	39/39	35.9 (14)	17.9 (7)	0.125 ‡
BMI	kg/m^2^	39/38	28.55 ± 5.09	28.44 ± 5.68	0.951 †
RR	/min	33/34	24 ± 5	23 ± 6	0.362 †
SpO_2_	%	37/37	92 ± 2.5	93 ± 2.5	0.119
Patient with dyspnea ^b^	%	6/13	92 ± 2.5	92 ± 3	0.487
Oxygen supplementation requirement	% (n)	38/39	37.8 (14)	62.2 (23)	0.084
Patients with dyspnea ^b^	% (n)	6/13	50 (3)	71.4 (10)	0.393
O_2_ -supplementation	L/min	36/35	1.5 (±2)	2 (±2)	0.071 †
Patients with dyspnea ^b^	L/min	6/13	2 ± 2.5	3 ± 3	0.610
Symptom onset to hospitalization	days	39/39	8 ± 4	7 ± 4	0.335
Radiological evidence	% (n)	39/39	100 (39)	94.9 (37)	0.152
Quality of life ^c^		26/20			
SF-12 physical score			31.39 ± 10.61	27.85 ± 11.75	0.352 †
SF-12 mental score			41.76 ± 15.51	51.66 ± 8.04	0.018 †
Hematology					
Hemoglobin	g/L	30/34	129.67 ± 15.89	134.68 ± 15.78	0.211
RBC	10^6^/µL	31/34	4.35 ± 0.43	4.353 ± 0.74	0.440 †
Leukocytes	10^3^/µL	31/34	7.11 ± 2.46	7.647 ± 3.23	0.451
Thrombocytes	10^3^/µL	30/34	244.90 ± 81.02	238.56 ± 101.88	0.783
Blood chemistry					
25(OH)D3	nmol/L	37/39	31.46 ± 10.95	28.54 ± 10.13	0.232
PTH	pmol/L	7/6	3.14 ± 1.34	4.160 ± 3.93	0.943 †
Calcium	mmol/L	9/13	2.15 ± 0.06	2.09 ± 0.34	0.282 †
Phosphate	mmol/L	7/12	1.00 ± 0.29	0.86 ± 0.27	0.570
CRP ^d^	mg/L	28/31	96.50 ± 57.07	66.67 ± 61.23	0.015 †

Abbreviations: ALP = alkaline phosphatase, ALT = alanine aminotransferase, AST = aspartate aminotransferase, BMI = body mass index, BP = blood pressure, CRP = C-reactive protein, eGFR = estimated glomular filtration rate, MCV = mean corpuscular volume, SpO_2_ = peripheral oxygen saturation, RBC = red blood cells, RR = respiratory rate. Continuous variables: mean ± standard deviation (SD). Categorical variables: percentages of patients (absolute number of patients per group = n). ^a^ number of patients with available information. ^b^ Values for patients with self-reported dyspnea. ^c^ Assessment of the quality of life started when the trial was already ongoing, therefore not all patients have a baseline value. ^d^ Mean values are outside the normal range. ‡ Fisher’s exact test when expected frequencies <5 instead of Chi^2^ test for categorical variables. † Mann–Whitney-U-Test.

**Table 3 biomedicines-11-01277-t003:** Length of hospital stay per group.

Length of Stay in Days	Intervention, n = 39	Control, n = 39	*p*-Value
	Median (IQR)	Median (IQR)	
Time from symptom onset to discharge	16 (9)	15 (8)	0.193 †
Time from hospital admission to discharge	8 (8)	8 (6)	0.924 †
Time from randomization to discharge	6 (10)	6 (6)	0.920 †

† Mann–Whitney-U-Test.

**Table 4 biomedicines-11-01277-t004:** Length of hospital stay in patients with severe vitamin D deficiency.

Length of Stay (Days) When Serum 25(OH) D Level <25 nmol /L, n = 25	Intervention, n = 12	Control, n = 13	*p*-Value
	Median (IQR)	Median (IQR)	
Time from symptom onset to discharge	15 (11)	15 (9)	0.764 †
Time from hospital admission to discharge	5.5 (13)	9 (6)	0.299 †
Time from randomization to discharge	4.5 (13)	7 (5)	0.444 †

† Mann–Whitney-U-Test.

**Table 5 biomedicines-11-01277-t005:** Odds ratio of COVID-19-related complications.

COVID-19-Related Complications	Odds Ratio	95% Confidence Interval		*p*-Value
		Lower Level	Upper Level	
ICU Admission	1.371	0.232	4.319	1.00
ICU Admission COVID-19-related ^a^	1.371	0.286	6.576	0.693
Intubation	4.343	0.463	40.749	0.199
Other complications	1.542	0.243	9.776	0.646
Death	0.487	0.042	5.601	0.564

^a^ One patient was excluded from the corrected analysis, as the reason for ICU admission was not COVID-19-related.

**Table 6 biomedicines-11-01277-t006:** Change of serum 25(OH)D from randomization to discharge.

Serum 25(OH)D Level in nmol/L	N Available (Intervention/Control) ^a^	Intervention	Control	*p*-Value
25(OH)D level at randomization	37/39	31.46 ± 10.948 [11–44]	28.54 ± 10.13 [9–45]	0.232
25(OH)D level at discharge	21/26	50.95 ± 16.98 [2–83]	23.62 ± 8.88 [2–61]	<0.001 ***
Change 25(OH)D level	21/26	26.35 ± 8.88 [9–45]	−2.73 ± 10.23 [−33–16]	<0.001 ***
Patients with 25(OH)D level >50 nmol at discharge	21/26	23.1 (9)	2.6 (1)	0.003 ‡ **

Continuous variables: mean ± standard deviation (SD) [range]. Categorical variables: percentages of patients (absolute number of patients per group = n). ^a^ number of patients with available information. ‡ Fisher’s exact test when expected frequencies <5 instead of Chi^2^ test for categorical variables. Level of significance: ** < 0.01 *** < 0.001.

## Data Availability

The data presented in this study are available on request from the corresponding author. The data are not publicly available due to privacy policy and ethical statement.

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
