# Peer review of "Effect of Single High Dose Vitamin D Substitution in Hospitalized COVID-19 Patients with Vitamin D Deficiency on Length of Hospital Stay"

_biomedicines, 2023, doi:10.3390/biomedicines11051277_

Round 1

Reviewer 1 Report

 Thank you very much to the Editor of BIOMEDICINES for allowing me to review the paper entitled ‘Effect of Single High Dose Vitamin D substitution in hospitalized Covid-19 patients with Vitamin D deficiency on length of hospital stay.’

Major comments:

The authors suggested that a single intervention with 140 000 IU (3500 μg) vitamin D and treatment, as usual, did not significantly shorten the length of hospital stay (in the group of COVID-19 patients with laboratory-confirmed serum 25-hydroxyvitamin D < 50nmol/l) but was effective and safe for the elevation of serum 25(OH)-vitamin D levels. Moreover, the subgroup analysis in patients with severe 25-hydroxyvitamin D-deficiency (< 25nmol/l) there are showed a non-significant reduction in the median length of hospital stay in the intervention group (5.5 vs 9 days). Results were given clearly with sufficient tables, figures and data analysis, but in some figures, the font is too hard to read (Figure 1A, Figure 1B, Figure 1C, Figure 2A, Figure 2B, Figure 2C).

Table 1 appears twice in the article with different data.

Moreover, in paper lack of comments in the Discussion about differences in CRP levels (mg/L 96.50 ± 57.07 66.67 ± 61.23, p=0.015) and SF-12 mental score (41.76 ± 15.51 51.66 ± 8.04 p=0.018).

Studies are based on relatively new literature (8 out of 54 items were published more than five years ago).

Please verify the entry:

All patients that were hospitalized due to Covid-19 in one of the participating centers (Cantonal hospital Baselland and Cantonal hospital St. Gallen) were considered as potential eligible for participation in this trial and therefore checked for the inclusion and exclusion criteria (Error! Reference source not found.).

Please verify the entry:

In the competing risks model with the full dataset and death as the competing risk, groups did not differ in a) time from symptom onset to discharge (HR= 0.88, 95% CI 0.57 – 1.36, p= 0.570) b) time from hospital admission to discharge (HR= 0.99, 95% CI 0.64 – 1.52, p= 0.950) c) time from randomization to discharge (HR= 0.96, 95% CI 0.62 – 1.48, p= 0.850) Error! Reference source not found.Figure 2)

Please verify the entry:

As one patient was not admitted due to Covid-19, we also performed a corrected comparison, which included only patients, who were admitted to ICU due to Covid-19 (Error! Reference source not found.)

Minor comments:

Instead of: The subgroup anaylsis in patients with severe Vit-D-deficiency (< 25nmol/l) showed a non-significant reduction in the median lenghth of hospital stay in the intervention group (5.5 vs. 9 days, p=.299).

Should be: The subgroup analysis in patients with severe 25-hydroxyvitamin D -deficiency (< 25nmol/l) showed a non-significant reduction in the median length of hospital stay in the intervention group (5.5 vs 9 days, p=.299).

(IN TABLE 1) 

Instead of: Laboratory confirmed serum 25-hydroxvitamin D < 50nmol/l (< 20ng/ml)

Should be: Laboratory-confirmed serum 25-hydroxyvitamin D level < 50nmol/l (< 20ng/ml)

Instead of: The competing risk model with death did not reval seignificant differences between the group in the length of stay (HR= 0.96, 95% CI 0.62 – 1.48, p= 0.850). 

Should be: The competing risk model with death did not reveal significant differences between the group in the length of stay in the hospital (HR= 0.96, 95% CI: 0.62 – 1.48, p= 0.850). 

Instead of: To elevate the serum 25(OH)D level quickly and sufficiently, a single dose of 140`000 IU (3’500 μg) vitamin D3 was administered orally.

Should be: To elevate the serum 25(OH)D level quickly and sufficiently, a single dose of 140 000 IU (3 500 μg) vitamin D3 was administered orally.

Instead of: The intervention group had a significant higher mean CRP value at time of randomization compared to the control group (intervention: 96.50 mg/L vs. control: 66.67 mg/L, p =.015), no other value in the hematology profile or blood chemistry showed a significant difference between the groups.

Should be: The intervention group had a significantly higher mean CRP value at the time of randomization compared to the control group (intervention: 96.50 mg/L vs control: 66.67 mg/L, p =0.015); no other value in the hematology profile or blood chemistry showed a significant difference between the groups. 

Author Response

Dear Reviewer,

Thank you very much for taking the time and reviewing our manuscript. We really appreciated your valuable feedback and  comments.

For the point by point answer, please see the attached document. 

Reviewer 2 Report

Major comments:

1. It is obvious (and has been shown multiple times before) that a single vitamin D bolus can elevate the vitamin D status. Please discuss and cite references.

2. The style and labeling of the figures should be harmonized. Please have the labelling large enough to be easily readable. Moreover, the resolution of the figures needs to be improved. This applies also to the supplements.

3. Please discuss in more detail the negative outcome of the study. Could the fact that probably 25% of the patients are low vitamin D responders affect the outcome (see PMID: 28034764)?

Minor comments:

1. Please be throughout the whole manuscript more specific which vitamin D metabolite (vitamin D3, 25(OH)D3, 1,25(OH)2D3) is meant when you use the term "vitamin D". Moreover, please use always the same term for the same molecule.

2. Please be more accurate with abbreviations, i.e. define them at first time use and apply them then consistently. This applies also to the Abstract. Moreover, please use always the same abbreviation (e.g. Covid-19 versus COVID-19).

3. There are error messages concerning some references, please correct.

4. Please be consistent whether you have a space between number and unit (or not).

5. Please use units consistently, such as mol/l versus mol/L.

Author Response

Dear Reviewer, 

Thank you very much for taking the time and reviewing our manuscript. We really appreciated your comments.

Please see the attached document for the detailed point-by-point reply. 

Reviewer 3 Report

The manuscript evaluating the effect of high dose of vitamin D on hospital in COVID-19 patients stay is well written. The conclusion is supported by the data provided. The authors have mentioned the limitations of the study. Please check for typos in the text. COVID-19 should be consistent throughout the text. decimal values should follow same manner to write (0.658 or .658). The references are not as per journal style, please check. They should be numbered in succession but  (56,58–61, 62–66) are after 16.

Please mention Table 1 after exclusion and inclusion criteria.

Author Response

(The authors gave the same response as above.)

Round 2

Reviewer 2 Report

Minor comment:  Table 1 does not follow the consistency about units.